# Soluble IL-2R Levels at Baseline Predict the Development of Severe Respiratory Failure and Mortality in COVID-19 Patients

**DOI:** 10.3390/v14040787

**Published:** 2022-04-10

**Authors:** Nikolaos K. Gatselis, Vasiliki Lygoura, Aggeliki Lyberopoulou, George Giannoulis, Anna Samakidou, Antonia Vaiou, George Vatidis, Katerina Antoniou, Aggelos Stefos, Sarah Georgiadou, Dimitrios Sagris, Dafni Sveroni, Despoina Stergioula, Stella Gabeta, George Ntaios, George N. Dalekos

**Affiliations:** Department of Medicine and Research Laboratory of Internal Medicine, National Expertise Center of Greece in Autoimmune Liver Diseases, European Reference Network on Hepatological Diseases (ERN RARE-LIVER), General University Hospital of Larissa, 41110 Larissa, Greece; lvasiliki@med.uth.gr (V.L.); aliber@med.uth.gr (A.L.); geogiannoulis7@gmail.com (G.G.); anna__sam@hotmail.com (A.S.); tonia1194@hotmail.com (A.V.); vatgeo@hotmail.com (G.V.); kate.antoniou1@gmail.com (K.A.); agstefos@hotmail.com (A.S.); s_georgiadou@hotmail.com (S.G.); disagris@uth.gr (D.S.); amishdafni@hotmail.com (D.S.); deppyst@gmail.com (D.S.); s_gampeta@yahoo.com (S.G.); gntaios@med.uth.gr (G.N.); dalekos@uth.gr (G.N.D.)

**Keywords:** biomarker, coronavirus disease-19, soluble interleukin-2 receptor

## Abstract

Risk stratification of coronavirus disease-19 (COVID-19) patients by simple markers is critical to guide treatment. We studied the predictive value of soluble interleukin-2 receptor (sIL-2R) for the early identification of patients at risk of developing severe clinical outcomes. sIL-2R levels were measured in 197 patients (60.9% males; median age 61 years; moderate disease, *n* = 65; severe, *n* = 132, intubated and/or died, *n* = 42). All patients received combined immunotherapies (anakinra ± corticosteroids ± intravenous immunoglobulin ± tocilizumab) according to our local treatment algorithm. The endpoint was the composite event of intubation due to severe respiratory failure (SRF) or mortality. Median (interquartile range) sIL-2R levels were significantly higher in patients with severe disease, compared with those with moderate disease (6 (6.2) vs. 5.2 (3.4) ng/mL, *p* = 0.017). sIL-2R was the strongest laboratory predictive factor for intubation/death (hazard ratio 1.749, 95%CI 1.041–2.939, *p* = 0.035) after adjustment for other known risk factors. Youden’s index revealed optimal sIL-2R cut-off for predicting intubation/death at 9 ng/mL (sensitivity: 67%; specificity: 86%; positive and negative predictive value: 57% and 91%, respectively). Delta sIL-2R between the day of event or discharge minus admission date was higher in patients that intubated/died than in those who did not experience an event (2.91 (10.42) vs. 0.44 (2.88) ng/mL; *p* = 0.08)). sIL-2R on admission and its dynamic changes during follow-up may reflect disease severity and predict the development of SRF and mortality.

## 1. Introduction

Coronavirus disease 2019 (COVID-19) has a wide clinical spectrum, ranging from asymptomatic to severe respiratory failure (SRF), multiorgan failure, and death [1,2,3,4]. Indeed, a substantial proportion of patients with COVID-19 develop severe pneumonia that requires hospitalisation [5] and may progress to critical illness with SRF requiring invasive mechanical ventilation [6,7].

Evidence suggests that the pathophysiology of COVID-19 deterioration is based on immune dysregulation and hyperinflammation resembling cytokine release syndrome, driven by several pro-inflammatory cytokines [8,9]. Excessive and dysregulated immune responses lead to disproportionate inflammation and harmful tissue damage both locally and systematically [10]. Under this context, early identification of biomarkers for disease progression is critical for guiding therapeutic approaches [11,12,13,14]. High serum cytokines, such as interleukin-6 (IL-6), inflammatory serum markers, such as ferritin, D-dimer, C-reactive protein (CRP), and peripheral lymphopenia are some characteristics of immunological signatures of COVID-19 [1,15].

However, to date, few studies have investigated the correlation between COVID-19 severity and soluble interleukin-2 receptor (sIL-2R), which is one of the biomarkers of immunological dysregulation [16,17,18,19,20,21,22]. Interleukin-2 (IL-2) interacts with its receptors on target cells to activate and maintain immune responses. IL-2 receptors appear on the cell surface of activated T cells, and its α-chain (CD25), an IL-2-receptor component, is released into the serum through proteolytic cleavage, known as sIL-2R [23,24]. We and others have shown that disorders associated with cellular activation, such as autoimmune, neoplastic, and infectious diseases, are characterised by increased serum sIL-2R concentrations, which correlate with disease activity [25,26,27,28,29].

Accordingly, the aim of the present study was to determine the predictive value of sIL-2R levels for identifying COVID-19 patients who are at risk of deterioration to severe clinical outcomes as timely initiation of targeted interventions in these patients seems crucial to prevent progression to SRF and reduce mortality.

## 2. Materials and Methods

sIL-2R levels were measured in the sera of 197 patients (60.9% males; median age 61 years) hospitalised in our Department due to moderate or severe pneumonia caused by the severe acute respiratory syndrome coronavirus 2 (SARS-CoV-2). Moderate pneumonia was defined by oxygen saturation by pulse oximetry in room air (SpO*2*) ≥ 94%, whereas severe pneumonia was characterised by the presence of at least one of the following: SpO*2* < 94% in room air, breath rate > 30/min, respiratory ratio (pO*2*/FiO*2*) < 300, and pulmonary infiltrates in more than >50% of the lung parenchyma [1].

Demographics (age, sex, body mass index (BMI), smoking habits), comorbidities (presence of diabetes, chronic obstructive pulmonary disease (COPD), cardiovascular diseases), parameters of disease severity (pO*2*/FiO*2* ratio, respiratory rate), and laboratory inflammation markers (lymphocytes, ferritin, CRP) were recorded at baseline. All patients were treated with combined immunotherapies (anakinra ± corticosteroids ± intravenous immunoglobulin ± tocilizumab) based on initial ferritin levels and according to our local treatment algorithm [1].

sIL-2R serum levels were determined by using sIL-2 Receptor Human ELISA (BioVendor Research and Diagnostic Products^®^, Brno, Czech Republic), according to the manufacturer’s instructions. Serum samples of patients were collected and stored frozen at −20 °C until use. Samples were thawed only once to prevent repeated freeze–thaw cycles. A standard curve was constructed using 7 standards of twofold serial dilutions from 20 ng/mL to 0.31 ng/mL. To avoid a Hook effect, samples with a concentration greater than the standard of 20 ng/mL were further diluted to precisely quantitate the actual sIL-2R level. The calculated overall intra-assay and inter-assay coefficients of variations were 7.2% and 9.8%, respectively.

The composite endpoint of our study was the time from hospital admission to intubation due to SRF or death due to COVID-19. SRF was defined as the presence of a decrease in pO*2*/FiO*2* ratio lower than 150 and the need for mechanical ventilation. For patients who died after intubation, the timing of the composite endpoint was defined at the time of intubation. Patients without an endpoint event had their data censored after 30 days of follow-up.

All patients consented to participate in this study. The Ethics Committee of Research of Thessaly University approved the study protocol, which conforms to the guidelines of the 1975 Declaration of Helsinki, revised in Brazil in 2013, as reflected in a priori approval by the institution’s human research committee.

### Statistical Analysis

The Kolmogorov–Smirnov test was applied to assess the normal distribution of variables, and quantitative values were expressed as median (interquartile range (IQR)). Mann–Whitney U-test, x^2^ (two-by-two with Yate’s correction), Fisher’s exact test, Spearman’s rho correlation, and binary logistic regression analysis were used to analyse the data. The impact of baseline clinical and laboratory risk factors on our composite endpoint of intubation/mortality was evaluated by univariate and multivariate Cox regression analyses. Receiver operating characteristic (ROC) curves were constructed, and comparisons were performed by the DeLong test. Youden indices (sensitivity + specificity − 1) were calculated to establish optimal cut-offs for sIL-2R. Two-sided *p*-values less than 0.05 were considered statistically significant. All data analyses were carried out with SPSS version 26.0.

## 3. Results

### 3.1. Demographic, Clinical, and Laboratory Characteristics

Among the 197 patients (120 males and 77 females) with COVID-19 recruited in this study, 92 had moderate and 105 had severe disease at the time of first evaluation. Overall, 27 patients with moderate disease (29.3%) progressed to severe disease during hospitalisation.

The demographic, clinical, and laboratory characteristics of patients are summarised in Table 1. The median (IQR) age of patients was 61 (21) years. The most common comorbidity was cardiovascular diseases (104/197, 52.8%), followed by diabetes (38/197, 19.3%). The median (IQR) duration of symptomatic disease till admission was 7 (5) days.

Patients with severe disease were characterised by older age (*p* < 0.001), increased BMI (*p* = 0.018), higher prevalence of cardiovascular disease (*p* = 0.001), lower lymphocytes (*p* = 0.005), and higher CRP (*p* < 0.001) and ferritin (*p* < 0.001) levels at the time of presentation.

sIL-2R levels were significantly higher in patients with severe disease (6 (6.2) vs. 5.2 (3.4) ng/mL, *p* = 0.017). sIL-2R levels were negatively correlated with respiratory ratio (r = −0.335; *p* < 0.001) and lymphocytes count (r = −0.192, *p* = 0.007). A positive correlation was found between sIL-2R and CRP (r = 0.253; *p* < 0.001), whereas no correlation was found with ferritin (r = 0.111; *p* = 0.123).

### 3.2. Outcome of Patients

During follow-up, 42/197 (21.3%) were intubated or died. Table 2 shows the baseline factors associated with intubation/mortality. 

Univariate analysis revealed that intubation/mortality was associated with advanced age (hazard ratio, HR = 1.056, 95% confidence interval (CI): 1.031–1.081, *p* < 0.001); presence of diabetes (HR = 1.983, 95%CI: 1.031–3.815); cardiovascular disease (HR = 3.222, 95%CI: 1.583–6.557, *p* = 0.001); lower pO*2*/FiO*2* ratio (HR = 0.987, 95%CI: 0.984–0.990, *p* < 0.001); decreased lymphocytes count (HR = 0.428, 95%CI: 0.238–0.768, *p* = 0.004); increased ferritin (HR = 1.374, 95%CI 1.035–1.824, *p* = 0.028); CRP (HR = 1.569, 95%CI: 1.209–2.035, *p* = 0.001); sIL-2R levels (HR = 3.251, 95%CI: 2.171–4.869, *p* < 0.001).

However, in the multivariate analysis, the strongest negative predictive clinical markers were older age (HR = 1.036, 95%CI: 1.004–1.070, *p* = 0.030) and low pO*2*/FiO*2* ratio (HR = 0.985, 95%CI: 0.979–0.990, *p* < 0.001), while only the increased sIL-2R levels maintained a significant negative prediction for intubation or death among the laboratory markers (HR = 1.749, 95%: 1.041–2.939, *p* = 0.035; Table 2).

The AUC (95%CI) of sIL-2R for detecting patients with COVID-19 who eventually intubated or died was 0.789 (0.709–0.869). sIL-2R at presentation had higher predictive ability when compared with ferritin (AUC (95%CI): 0.612 (0.513–0.711); *p* = 0.004), while a trend was found when compared with CRP levels at baseline (AUC (95%CI): 0.683 (0.594–0.773); *p* = 0.065) (Figure 1).

Youden’s index revealed an optimal cut-off of sIL-2R for predicting intubation or death at 9 ng/mL with sensitivity of 67% and specificity of 86% (PPV: 86%, NPV: 91%). Patients with sIL-2R ≥ 9 ng/mL were characterised by increased age (*p* < 0.001), lower pO*2*/FiO*2* ratio (*p* < 0.001), increased respiratory rate (*p* = 0.008), lower lymphocytes (*p* = 0.007) and higher CRP levels (*p* = 0.003; Table 3).

### 3.3. sIL-2R, Ferritin, CRP, and Lymphocytes Levels Alteration during Hospitalisation and Outcome

sIL-2R, ferritin, CRP, and lymphocytes levels were measured in serial available samples of 23 patients (7 with moderate and 16 with severe disease). Serum samples were collected at baseline, in the middle of hospitalisation, and at discharge or at the time point of intubation/death. During hospitalisation, 9/23 patients were intubated or died (all of them from the 16 patients in the group with severe disease).

A comparison of median (IQR) sIL-2R at three different time points between patients who were intubated or died (*n* = 9) and patients who were discharged from the hospital without intubation (*n* = 14) showed higher levels for the former group at three different time points: baseline (10.1 (4) vs. 3.7 (2.8) ng/mL; *p* = 0.001), middle of hospitalisation (9.4 (4.2) vs. 4.9 (4.2) ng/mL; *p* = 0.002), time of discharge or intubation/death (14.6 (14.2) vs. 3.9 (4) ng/mL; *p* = 0.002) (Figure 2A). Furthermore, patients who were intubated or died had higher ferritin levels (1360 (1352) vs. 455 (583) ng/mL; *p* = 0.013) (Figure 2B) and lower lymphocytes levels (590 (840) vs. 1925 (1500) /μL; *p* = 0.001) (Figure 2D) only at the time of intubation/death, compared with patients who were eventually discharged, while no difference was found at middle of hospitalisation (ferritin: 1294 (1707) vs. 653 (704) ng/mL; *p* = 0.109 and lymphocytes: 700 (1040) vs. 845 (670); *p* = 0.477) and baseline between the two groups of the patients (ferritin: 634 (1862) vs. 458 (533); *p* = 0.141 and lymphocytes: 900 (525) vs. 1075 (833); *p* = 0.250) (Figure 2B,D). No differences were also identified between the two groups, regarding CRP levels at three time points (baseline: 6.7 (13) vs. 3.8 (7.2); *p* = 0.159, middle of hospitalisation: 1.5 (2.9) vs. 0.9 (1.1), *p* = 0.109, time of intubation/death or discharge: 0.2 (0.7) vs. 0.1 (0.2); *p* = 0.083) (Figure 2C).

Median (IQR) delta sIL-2R (day of event or discharge day of admission) was higher in the group of patients that intubated or died (2.91 (10.42) ng/mL), compared with patients who did not experience an event (0.44 (2.88) ng/mL; *p* = 0.08)) (Figure 2A). In addition, patients that did not reach the endpoint of intubation/death had higher median (IQR) delta of lymphocytes during hospitalisation (1120 (1685) vs. 110 (420) /μL; *p* = 0.011) (Figure 2D), while no difference was found regarding delta ferritin (*p* = 0.781, Figure 2B) and delta CRP (*p* = 0.336, Figure 2C).

## 4. Discussion

In the current study, we evaluated the relationship between sIL-2R levels and the disease severity of COVID-19 pneumonia and outcomes of patients. Our analysis demonstrated that sIL-2R was associated with severe pneumonia and mainly that sIL-2R levels at baseline were the strongest laboratory-negative predictive factor for patient outcome. According to our results, a level of sIL-2R greater than 9 ng/mL had a PPV of 86% and an NPV of 91% for intubation or death.

From the first days of the pandemic, serum concentrations of both anti-inflammatory (IL-10) and pro-inflammatory cytokines (IL-2, IL-6, TNF-α) were found to be increased in severe cases of COVID-19, compared with patients with moderate disease, implying that a cytokine storm is linked to disease pathogenesis [15]. However, as we recently reported [1], stratifying patients based on the presence of these proinflammatory cytokines may be premature due to contradictory data regarding their ability to predict disease severity and outcome, although they are found to be increased in serum [30,31,32]. Therefore, in our study, we sought to investigate the utility of sIL-2R in COVID-19, a much more specific marker of disease activity in diseases characterised by immune dysregulation (cancer, autoimmune disorders, lymphomas, ulcerative colitis, and infections) [25,26,27,28,29].

Moreover, several laboratory markers have been thoroughly investigated in an attempt to develop early diagnostic and prognostic biomarkers for predicting COVID-19 progression to severe disease and outcome. In this context, hyperferritinaemia, high CRP levels, and lymphopenia, along with some more sophisticated markers such as the soluble urokinase plasminogen activator receptor (suPAR) ≥ 6 ng/mL and SCOPE score ≥ 6, have already been associated with severity and survival of COVID-19 patients [6,7,14,33,34,35,36]. However, our multivariate analysis showed that sIL-2R had the highest diagnostic accuracy for predicting intubation or death, compared with the other known negative predictive factors (ferritin, CRP, and lymphopenia). In addition, rapid suPAR testing is frequently not available in the routine hospital setting, while the SCOPE score needs the simultaneous determination of four indices—namely, CRP, ferritin, IL-6, and D-dimer.

Notably, in parallel with other reports, we found that sIL-2R levels are negatively associated with lymphocytes count, as has also been shown in previous studies [20,22]. Several theories have been proposed to explain the mechanisms of lymphopenia in COVID-19 patients [2,37], such as extensive lymphocytes infiltration in affected organs [38], infection of lymphocytes caused by viral attachment to their ACE2 receptor, resulting in lymphocyte death [39], and induction of lymphocyte apoptosis caused by pro-inflammatory cytokines [20,40,41]. Many subsets of T cells (e.g., Tregs, CD4+, and CD8+ cells), require IL-2 for their proliferation, differentiation, and function [39]. In the case of COVID-19, it has been proposed that circulating sIL-2R causes lymphopenia by blocking IL-2 signalling [20]. In particular, correlation analysis between sIL-2R concentrations and the proportion of different immune cell types in COVID-19 patients suggested that sIL-2R may be a negative regulatory factor for T cells, particularly CD8+ T cells, but not CD4+ T cells, NK cells, or B cells [20].

Regarding the outcome of patients, our data showed that increased sIL-2R levels on admission are associated with increased rates of intubation or mortality. In line with our findings, previous studies have shown that sIL-2R levels are associated with worst outcomes, as attested by longer duration of illness and time to recover, and lower survival rates [16,17,18]. Most importantly, however, in our study, we were able to show that a cut-off level of 9 ng/mL could be used as a predictive marker for identifying patients who are at risk for intubation/death with high diagnostic accuracy.

Moreover, sequential monitoring of sIL2R may add further diagnostic value for patients who are likely to bear poor prognoses. Indeed, according to our subanalysis, individuals who were intubated or died had consistently higher levels of sIL-2R during hospitalisation, compared with patients who did not experience an event, while ferritin and lymphocytes differ only at the time point of the event, which attenuates their predictive values. Moreover, the increase in sIL-2R levels was higher in patients with an unfavourable course of the disease.

Indeed, future research is needed to evaluate the relationship between sIL-2R serum levels and immunological response to various stimuli in COVID-19 patients; however, the cross-sectional design of our study did not allow us to conduct such tests.

## 5. Conclusions

In conclusion, the simple determination of sIL-2R on admission by an easy-to-perform, low-cost, non-invasive test method that can be performed in any laboratory, as well as its sequential measurements during the disease course, seem to reflect COVID-19 severity and predict the development of SRF and death. As a result, it could be utilised as a biomarker in risk stratification of the patients in an attempt to promptly identify those who need intensive monitoring, prompt initiation of treatment, and/or inclusion in therapeutic trials.

## Figures and Tables

**Figure 1 viruses-14-00787-f001:**
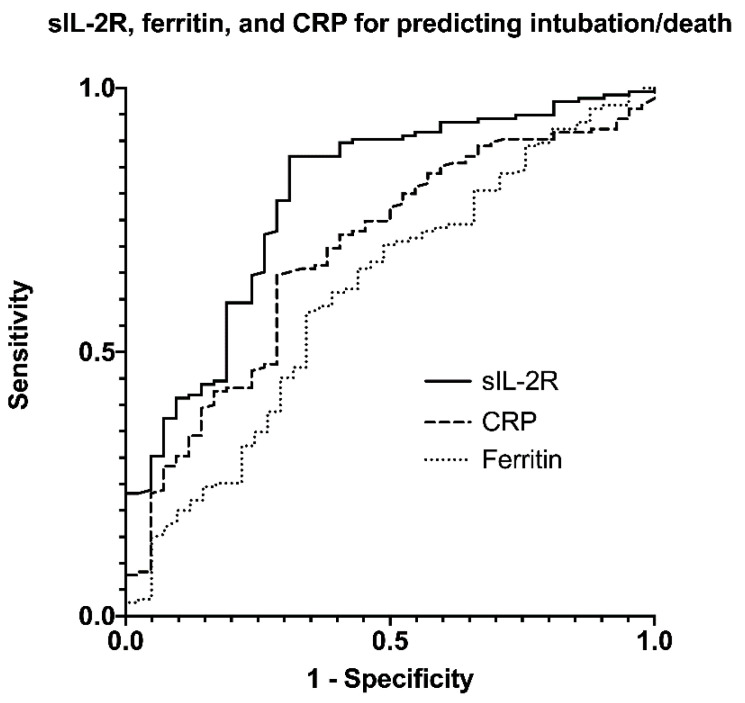
Receiver operating characteristic curves for the prediction of intubation or death according to sIL-2R, ferritin and CRP. sIL-2R had the highest diagnostic accuracy for intubation/death (AUC (95% CI): 0.789 (0.709–0.869)), compared with ferritin (AUC (95%CI): 0.612 (0.513–0.711); *p* = 0.004) and CRP (AUC (95%CI): 0.683 (0.594–0.773); *p* = 0.065). sIL-2R: soluble interleukin-2 receptor; CRP: C-reactive protein; AUC: area under the curve.

**Figure 2 viruses-14-00787-f002:**
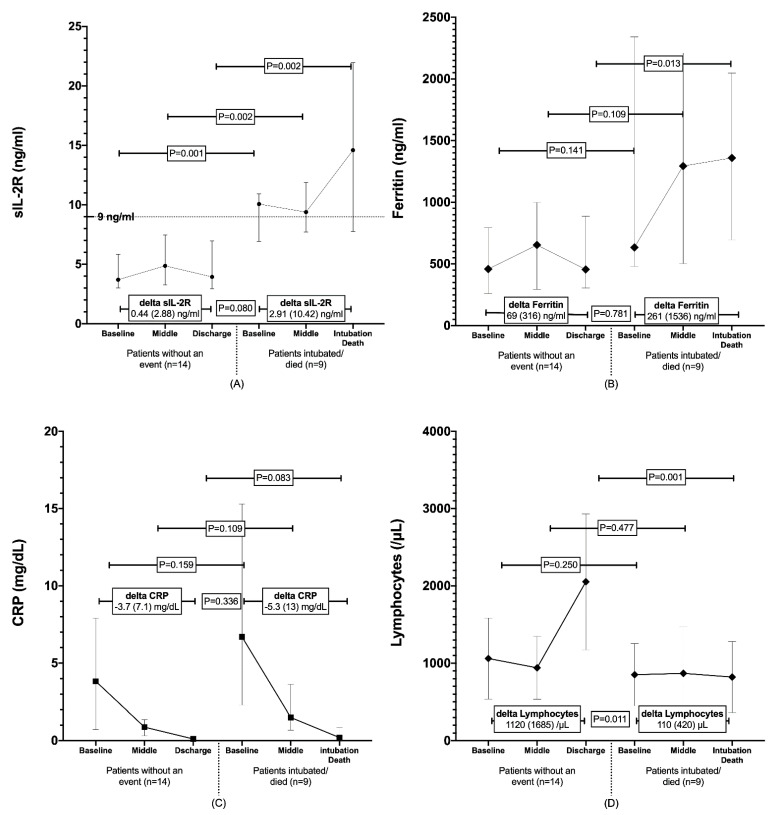
Serial changes in sIL-2R (**A**), ferritin (**B**), CRP (**C**), and lymphocyte (**D**) levels in 23 patients. (**A**) Patients intubated or died (*n* = 9) had higher levels of sIL-2R than patients without an event (*n* = 14) at baseline (10.1 (4) vs. 3.7 (2.8) ng/mL; *p* = 0.001), at middle of hospitalisation (9.4 (4.2) vs. 4.9 (4.2) ng/mL; *p* = 0.002), at the time of discharge or intubation/death (14.6 (14.2) vs. 3.9 (4) ng/mL; *p* = 0.002). Median (interquartile range) delta sIL-2R was higher in the group of patients that intubated or died (2.91 (10.42) vs. 0.44 (2.88) ng/mL; *p* = 0.08)). (**B**) Patients were intubated or died had higher ferritin levels (1360 (1352) vs. 455 (583) ng/mL; *p* = 0.013) only at the time of intuba-tion/death. (**C**) No differences were identified between the two groups, regarding CRP levels at three time points. (**D**) Patients intubated or died had lower lymphocytes levels (590 (840) vs. 1925 (1500) /μL; *p* = 0.001) only at the time of intubation/death and lower median (IQR) delta of lym-phocytes during hospitalisation (110 (420) vs 1120 (1685) /μL; *p* = 0.011). sIL-2R: soluble interleu-kin-2 receptor; CRP: C-reactive protein. sIL-2R: soluble interleukin-2 receptor; CRP: C-reactive protein.

**Table 1 viruses-14-00787-t001:** Baseline characteristics according to disease severity.

	Total (*n* = 197)	Moderate (*n* = 65)	Severe (*n* = 132)	*p*-Value
Age, median (IQR), years	61 (21)	57 (20)	65 (19)	<0.001
Male sex, *n* (%)	120 (60.9%)	37 (56.9%)	83 (62.9%)	0.516
BMI, median (IQR), kg/m^2^	27.7 (6)	26.4 (5.7)	27.8 (6.2)	0.018
Diabetes, *n* (%)	38 (19.3%)	8 (12.3%)	30 (22.7%)	0.121
COPD, *n* (%)	16 (8.1%)	3 (4.6%)	13 (9.8%)	0.440
Cardiovascular disease, *n* (%)	104 (52.8%)	20 (30.8%)	84 (63.6%)	0.001
Smoking, *n* (%)	73 (37.1%)	24 (36.9%)	49 (37.1%)	1.000
Disease duration, median (IQR), days	7 (5)	7 (6)	7 (4)	0.392
pO*2*/FiO*2* ratio, median (IQR)	304 (127)	376 (91)	252 (77)	<0.001
Respiratory rate, median (IQR), /min	22 (10)	20 (6)	24 (10)	<0.001
Lymphocytes, median (IQR), /μL	950 (595)	1050 (495)	885 (633)	0.005
Ferritin, median (IQR), ng/mL	477 (800)	290 (324)	591 (938)	<0.001
CRP, median (IQR), mg/dL	3.5 (8.2)	1.7 (3.7)	4.7 (9.9)	<0.001
sIL-2R, median (IQR), ng/mL	5.8 (5.2)	5.2 (3.4)	6 (6.2)	0.017

Abbreviations are same as in text. IQR: interquartile range. BMI: body mass index. COPD: chronic obstructive pulmonary disease. CRP: C-reactive protein. sIL-2R: soluble interleukin-2 receptor.

**Table 2 viruses-14-00787-t002:** Baseline factors associated with intubation/mortality.

	Univariate Analysis			Multivariate Analysis		
	HR	95% CI	*p*	HR	95% CI	*p*
Age	1.056	1.031–1.081	<0.001	1.036	1.004–1.070	0.030
Male sex	1.339	0.705–2.544	0.372			
BMI	1.022	0.966–1.080	0.453			
Diabetes	1.983	1.031–3.815	0.040	0.562	0.256–1.238	0.153
COPD	0.536	0.130–2.218	0.390			
Cardiovascular disease	3.222	1.583–6.557	0.001	1.706	0.745–3.903	0.206
Smoking (previous/active)	1.179	0.637–2.183	0.601			
Disease duration	1.024	0.941–1.114	0.576			
pO_2_/FiO_2_ ratio	0.987	0.984–0.990	<0.001	0.985	0.979–0.990	<0.001
Lymphocytes	0.428	0.238–0.768	0.004	0.746	0.387–1.438	0.382
Ferritin	1.374	1.035–1.824	0.028	0.833	0.584–1.188	0.312
CRP	1.569	1.209–2.035	0.001	0.730	0.509–1.048	0.088
sIL-2R	3.251	2.171–4.869	<0.001	1.749	1.041–2.939	0.035

Abbreviations are same as in text. HR: hazard ratio; CI: confidence interval; BMI: body mass index; COPD: chronic obstructive pulmonary disease; CRP: C-reactive protein; sIL-2R: soluble interleukin-2 receptor.

**Table 3 viruses-14-00787-t003:** Baseline factors associated with increased sIL-2R levels (≥9 ng/mL).

	Total (*n* = 197)	sIL-2R ≥ 9 ng/mL (*n* = 50)	sIL-2R < 9 ng/mL (*n* = 147)	*p*-Value
Age, median (IQR), years	61 (21)	70 (16)	59 (21)	<0.001
Male sex, *n* (%)	120 (60.9%)	18 (36%)	59 (40.1%)	0.726
BMI, median (IQR), kg/m^2^	27.7 (6)	27.9 (5.4)	27.7 (6.2)	0.873
Diabetes, *n* (%)	38 (19.3%)	8 (16%)	30 (20.4%)	0.635
COPD, *n* (%)	16 (8.1%)	5 (10%)	11 (7.5%)	0.558
Cardiovascular disease, *n* (%)	104 (52.8%)	32 (64%)	72 (49%)	0.094
Smoking, *n* (%)	73 (37.1%)	20 (40%)	53 (36.1%)	0.742
Disease duration, median (IQR), days	7 (5)	7 (4)	7 (5)	0.391
pO*2*/FiO*2* ratio, median (IQR)	304 (127)	238 (147)	328 (119)	<0.001
Respiratory rate, median (IQR), /min	22 (10)	26 (10)	22 (8)	0.008
Lymphocytes, median (IQR), /μL	950 (595)	800 (453)	990 (640)	0.007
Ferritin, median (IQR), ng/mL	477 (800)	547 (1261)	454 (644)	0.094
CRP, median (IQR), mg/dL	3.5 (8.2)	6.8 (12.4)	2.6 (6.8)	0.003

Abbreviations are same as in text. sIL-2R: IQR: interquartile range; BMI: body mass index; COPD: chronic obstructive pulmonary disease; CRP: C-reactive protein.

## Data Availability

The data presented in this study are available on request from the corresponding authors.

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
