# Peer review of "Soluble IL-2R Levels at Baseline Predict the Development of Severe Respiratory Failure and Mortality in COVID-19 Patients"

_viruses, 2022, doi:10.3390/v14040787_

Round 1

Reviewer 1 Report

In the manuscript "Soluble IL-2R levels at baseline predict the development of severe respiratory failure and mortality in COVID-19 patients", the authors reported that median (interquartile range) sIL-2R were significantly higher in patients with severe COVID-19 compared to those with moderate COVID-19, sIL-2R was the strongest laboratory predictive factor for intubation/death after adjustment for other known risk factors, and 9 ng/mL was an optimal sIL-2R concentration for predicting COVID-19 patient intubation/death. These results are interesting and may benefit COVID-19 treatment by predicting COVID-19 severity, development of SRF and mortality. Below are my suggestion:

  1. By multivariate analysis, the authors showed that increased sIL-2R level is the only laboratory biomarker with a significant negative prediction for intubation or death. Data analysis is one of the most important part of this manuscript, which should be reviewed by a statistician.
  2. The author should also analyze the alterations of other laboratory markers (lymphocyte count, ferritin and CRP) during hospitalization and outcome using the same patient cohort and show the data in Figure 2.  Comparison of these laboratory marker alterations during hospitalization and outcome may corroborate the authors' claim that sIL-2R is the strongest laboratory biomarker for prediction of COVID-19 intubation or death.
  3. There is a typo in Figure 2. "Patient without and event" should be revised to "Patient without an event.

Author Response

REVIEWER #1:

  1. By multivariate analysis, the authors showed that increased sIL-2R level is the only laboratory biomarker with a significant negative prediction for intubation or death. Data analysis is one of the most important part of this manuscript, which should be reviewed by a statistician.

Response: We totally agree that with reviewer’s comment that statistical analysis is the most important part of our paper. For this reason, we perform a thorough and accurate analysis and further enhance our analysis after your thoughtful comment 2 (please see response to Reviewer’s 1, Comment 2). Please, consider that members of our group that have been extensively trained in biostatistics have performed extended evaluation and analysis of the results as always done in more than 250 of our international papers.

  1. The author should also analyze the alterations of other laboratory markers (lymphocyte count, ferritin, and CRP) during hospitalization and outcome using the same patient cohort and show the data in Figure 2.  Comparison of these laboratory marker alterations during hospitalization and outcome may corroborate the authors' claim that sIL-2R is the strongest laboratory biomarker for prediction of COVID-19 intubation or death.

Response: We thank the reviewer for the opportunity to further enhance our analyses and corroborate our finding that sIL-2R is the strongest laboratory biomarker for prediction of COVID-19 intubation or death. Following your suggestion, we revised the analysis presented in Figure 2, also including CRP, ferritin, and lymphocyte counts. Interestingly, sIL-2R was constantly higher (at baseline, in the middle of hospitalization, at the time of event) in the group of patients who experienced an event (intubation/death), whereas significantly increased ferritin and decreased lymphocytes levels were noticed only at the time of event (intubation/death), which attenuates their predictive value (please see revised Figure 2, revised Section 3.3 in page 6, revised Discussion in lines 253-256).

  1. There is a typo in Figure 2. "Patient without and event" should be revised to "Patient without an event.

Response:    The typo has been corrected in the revised Figure 2.

Reviewer 2 Report

This is a quite  interesting study which has demonstrated that sIL-2R serum secretion could become a predictive marker for  severe COVID_19. However , are missing some functional assays, looking for a negative correlation between  sIL-2R serum levels  and Immune response to mitogens, for example, at least in some patients; . Moreover the levels of IL-1b, IL-6, IL-2 and TNFa secretion should also be shown, to  investigate  a putative correlation between sIL2R and pro-inflammatory cytokine secretion levels 

Author Response

REVIEWER #2:

  1. This is a quite interesting study which has demonstrated that sIL-2R serum secretion could become a predictive marker for  severe COVID_19. However, are missing some functional assays, looking for a negative correlation between sIL-2R serum levels and Immune response to mitogens, for example, at least in some patients.

Response: We totally agree with Reviewer’s 2 comment that functional assays would further improve our results. However, the cross-sectional character of our study did not give us the opportunity to perform experiments to investigate the correlation of sIL-2R serum levels and immune response to factors (e.g., mitogens). A relevant comment has been added into “Discussion” as limitation of our study but also as a suggestion for future research (Please see Discussion, lines 257-259).  

  1. Moreover, the levels of IL-1b, IL-6, IL-2 and TNFa secretion should also be shown, to investigate a putative correlation between sIL2R and pro-inflammatory cytokine secretion levels.

Response: Thanks a lot for your comment. We absolute agree that cytokines (IL-1b, IL-6, IL-2 and TNFα) and their alterations are very important in understanding the pathogenesis of COVID-19 and therefore to improve our predictive ability regarding the clinical outcome of the disease. However, as we have also previously reported (Dalekos et al, Eur J Intern Med 2021, 88: 52-62), the stratification of patients according to the determination of these proinflammatory cytokines in serum could be premature due to contradictory results so far. Del Valle et al (Nat Med 2020, 26: 1636-43) showed after adjustment for disease severity, markers of inflammation, hypoxia, demographics, and several comorbidities, that IL-6 and TNF-α in serum were independent and significant predictors of disease severity and death. However, in other studies although the levels of these cytokines were elevated as compared to healthy, they were not associated with disease severity and survival (Aber et al, JCI Insight 2021, 6: e144455; Bel et al, iScience 2021, 24, 101896). Besides, they might be elevated at local sites of injury, and frequently do not correlate with the outcome following therapy with specific IL-1b or IL-6 inhibitors (Mehta, Lancet Rheumatol 2020; 2: e358-67). This is why, in our study, we sought to investigate the utility of sIL-2R in COVID-19, a much more specific marker of disease activity in diseases characterized by immune dysregulation (cancer, autoimmune disorders, lymphomas, ulcerative colitis and infections (Bien et al, Biomarkers 2008, 13, 1-26; Manoussakis et al, Clin Immunol Immunopathol 1989, 50, 321-332; Wagner et al, J Clin Oncol 1987, 5, 1262-1274; Dalekos et al, Gut 1993, 34, 658-664, Dalekos et al, Eur J Clin Invest 1993, 219-225). In order to satisfy you, we added a relevant comment in revised “Discussion, 2nd paragraph”, lines 212-218 and new References 30-32.
